# Toxic Effects of *Perilla frutescens* (L.) Britt. Essential Oil and Its Main Component on *Culex pipiens pallens* (Diptera: Culicidae)

**DOI:** 10.3390/plants12071516

**Published:** 2023-03-31

**Authors:** Ruimin Zhang, Wenxing Zhang, Junnan Zheng, Jingwei Xu, Huan Wang, Jiajia Du, Dan Zhou, Yan Sun, Bo Shen

**Affiliations:** Department of Pathogen Biology, Nanjing Medical University, Nanjing 211166, China

**Keywords:** essential oil, phytochemistry, traditional Chinese medicine, mosquito repellent insecticides, natural products

## Abstract

Sustainable control of mosquitoes, vectors of many pathogens and parasites, is a critical challenge. Chemical insecticides are gradually losing their effectiveness because of development of resistance, and plant metabolites are increasingly being recognized as potential alternatives to chemical insecticides. This study aimed to analyze the main components of *Perilla frutescens* essential oil (PE-EO), investigate the specific activity of PE-EO as a botanical insecticide and mosquito repellent, and explore whether its main constituents are potential candidates for further research. The larvicidal activity assay showed that LC50 of PE-EO and 2-hexanoylfuran was 45 and 25 mg/L, respectively. In the ovicidal activity assay, both 120 mg/L PE-EO and 80 mg/L 2-hexanoylfuran could achieve 98% egg mortality. Moreover, PE-EO and 2-hexanoylfuran showed repellency and oviposition deterrence effects. Notably, 10% PE-EO maintained a high rate of protection for 360 min. Although PE-EO and its main component had certain toxic effects on zebrafish, no significant harmful effects were detected in human embryonic kidney cells. Therefore, perilla essential oil is an effective agent for mosquito control at several life stages and that its main component, 2-hexanoylfuran, is a potential candidate for developing novel plant biopesticides.

## 1. Introduction

Mosquitoes are important disease vectors that carry various pathogens that cause diseases such as malaria, yellow fever, and Zika virus disease. According to the latest World Health Organization (WHO) data, there were 241 million cases of malaria worldwide in 2020, with an estimated 627 000 deaths [1]. Other data suggest that more than half of the global population is expected to be at risk of mosquito-borne disease transmission in 2050 [2].

To date, mosquito-borne disease control has relied primarily on the use of chemical insecticides [3]. However, this strategy is gradually losing its effectiveness. There are increasing reports that mosquitoes are rapidly developing resistance to common, WHO-approved chemical insecticides such as pyrethroids and organophosphates [4,5]. According to the WHO, the insecticide resistance in mosquitoes is a leading threat to mosquito-borne disease control and has a high potential to lead to renewed outbreaks of mosquito-borne diseases [6]. At the same time, the use of these pesticides causes environmental pollution and damages the ecosystem [7]. According to WHO estimates, 5000 deaths occur annually from organophosphorus pesticide poisoning.

Therefore, there is an urgent need to develop a novel type of safe and effective insecticide. Natural essential oils derived from plants provide a novel approach to mosquito vector control; they are usually safe for the environment, nontoxic to nontarget organisms, and leave minimal degradation residues in the field and water, minimizing ecological disruption [8]. Moreover, unlike chemical insecticides, essential oils exhibit multiple mechanisms of action owing to the diversity of active compounds in plant-derived extracts, which makes it difficult for mosquitoes to develop resistance to essential oils [9]. These various advantages suggest that essential oils can be developed as environmentally friendly biopesticides.

*Perilla frutescens*, a plant belonging to the Labiatae family, is widely distributed in East Asia, especially in China [10]. In Asia, perilla has long been used as a food and traditional medicine for various diseases such as colds, asthma, and allergies [11]. Wang Zhen has recorded its usage for arthropod control in *Nong Shu*, a text published during the Yuan Dynasty. Traditionally, people made a spice bag containing perilla and hung it over the body to repel insects. Recent studies report that perilla essential oil is harmful to *Aedes aegypti* larvae [12], and perilla essential oil is a critical component for long-term mosquito repellents [13]. Although perilla has long been used in China for its traditional applications, to our knowledge, no systematic studies have been conducted to determine its efficacy and effective chemical composition. The composition of the perilla essential oil is mostly detected as perilla ketone and perillaldehyde [14], but the composition and content of the essential oil components vary with the environment, and the detection of different novel substances has also been reported [15]. Therefore, this study aimed to analyze the components of perilla essential oil to assess their effectiveness in mosquito vector control and to elucidate the potential mechanism of action.

## 2. Results

### 2.1. Gas Chromatography–Mass Spectrometry Analysis of Perilla Essential Oil Composition

*Perilla frutescens* leaves were distilled by hydrodistillation to obtain an essential oil yield of 8 mL/kg. The chemical composition of the essential oil was assessed by comparing its fragmentation patterns in the mass spectra with those listed in the commercial NIST20 mass spectrometry library (MS). As shown in Figure 1 and Table 1, 30 fractions greater than 0.1% were identified, making up 94.2% of the essential oil. Among them, the main component was 2-hexanoylfuran (49.7%), which has the structural formula shown in Figure 1. The remaining noteworthy components included naphthalene, decahydro-1-methyl-2-methylene- (14.1%), β-caryophyllene (7.6%), and linalool (3.4%).

### 2.2. Larvicidal Activity

The 24 h acute toxicity of *Perilla frutescens* essential oil and its main component, 2-hexanoylfuran, to the third-instar larvae of *Cx. pipiens pallens* was positively correlated with the concentration. Although 65 mg/L of the essential oil caused 100% larval mortality, 2-hexanoylfuran was more toxic, with 45 mg/L causing 100% larval mortality. According to the assay, the concentration of the perilla essential oil that killed 50% of the larvae (LC50) was 45 mg/L, whereas the LC50 of 2-hexanoylfuran was 25 mg/L.

The morphology of larvae dead after the 24 h treatment was observed. Various degrees of darkening were found on the thorax and posterior side (Figure 2B), and such darkening was found in approximately 50% and 30% of dead larvae in the perilla essential oil and 2-hexanoylfuran groups, respectively, at LC50 Moreover, a few larvae were found to be completely darkened (Figure 2B(c)). Furthermore, varying degrees of muscle damage and loss of opacity were observed.

Small amounts of feed were added to the remaining surviving larvae in the LC50 group to observe the changes in growth and development. It was found that the feathered pupae also showed deformities such as direct damage to the tail and complete darkening (Figure 2B(f)). Among the remaining surviving larvae, 15% of such cases were observed in the 45 mg/L perilla essential oil group, and 12% were observed in the 25 mg/L 2-hexanoylfuran group. Approximately 7% appeared to be larval–pupal intermediates and pupal–adult intermediates (Figure 2B(g,h)). These intermediates died in the pupae, and no adult mosquitoes emerged from these pupae (Figure 2B(i)).

### 2.3. Ovicidal Activity

The hatching rates of mosquito eggs treated with different concentrations of perilla essential oil and 2-hexanoylfuran are presented in Table 2; it was found that the hatching rate of eggs was inversely proportional to the concentration of the treatment solution. When the concentration of essential oil reached 120 mg/L, most eggs did not hatch, reaching a mortality rate of 98.7%. The lethal effect of 2-hexanoylfuran on mosquito eggs was even higher, with this effect being achieved at 80 mg/L.

### 2.4. Repellency

Results showed that both perilla essential oil and 2-hexanoylfuran showed 100% mosquito repellency in the beginning, and the duration of protection increased with increasing concentration(Figure 3). The 10% (100,000 mg/L) concentration of perilla essential oil was the most effective, maintaining over 88% repellency at 360 min and 100% repellency during the first 60 min and maintaining parity with the positive control DEET at 330 min(Figure 3A). Additionally, 5% (50,000 mg/L) perilla essential oil showed moderate effect, showing 90% repellency at 60 min, which gradually decreased to 70% repellency. In contrast, 1% (10, 000 mg/L) perilla essential oil showed only approximately 52% repellency within 90 min. The duration of repellency was less than 30 min for both 1% and 5% 2-hexanoylfuran, whereas 10% 2-hexanoylfuran exhibited a repellency rate of 60–70% at 4 h(Figure 3B).

### 2.5. Oviposition Assay

Different concentrations of both perilla essential oil and 2-hexanoylfuran showed a repellency effect in the oviposition assay, and the effect was positively correlated with the concentration (Table 3). 2-hexanoylfuran was more effective in oviposition deterrence, reaching 93% effective repellency at 30 mg/L with an OAI of −0.88, whereas perilla essential oil showed 90% effective repellency at 100 mg/L with an OAI of −0.88.

### 2.6. Gut Histopathology

The third-instar larvae of *Cx. pipiens pallens* showed histological structural changes in the midgut epithelial cells after 24 h treatment with perilla essential oil and 2-hexanoylfuran at the LC50 and LC90 concentrations. In the control group, the midgut digestive cells showed regular distribution, had uniform cytoplasmic and nuclear markings, and had a more complete apical brush border. The peritrophic membrane and brush border of the midgut cells in the perilla essential oil treatment group were ruptured, with swollen nuclei, damaged cytoplasm, and deformed epithelial cells extending into the lumen (Figure 4C,D). In addition, the midgut cells in the 2-hexanoylfuran group were severely vacuolated, with marginalized nuclear chromatin and a severely diseased state (Figure 4E,F).

### 2.7. Effects on Nontarget Species

Nontarget toxicity assays showed that perilla essential oil and 2-hexanoylfuran were moderately toxic to zebrafish but not to human cells. Zebrafish treated with 45 mg/L (LC50) and 58 mg/L (LC90) perilla essential oil showed 43% and 76% mortality, respectively, whereas zebrafish treated with 25 mg/L (LC50) and 35 mg/L(LC90) of 2-hexanoylfuran showed 50% and 83% mortality, respectively. The CCK8 assay of 293T cells (Appendix A) showed no difference in cell viability between groups treated with perilla essential oil and 2-hexanoylfuran at LC50 and LC90.

## 3. Discussion

The essential oil derived from perilla, a herb traditionally used in Asia, is a light-yellow liquid with a strong and pleasant fragrance. Each kilogram of fresh leaves distilled using steam yields 8 mL of essential oil. Similar to other members in the family Labiatae, Perilla is rich in essential oil and has an advantage in yield [16]. The GC-MS analysis indicated that 2-hexanoylfuran was the major component (49.7%) of Perilla essential oil. The determination of 2-hexanoylfuran in the composition of perilla essential oil has been reported before, but the results showed that it was not the major component [15].

Larval source management is one of the main forms of vector control owing to the difficulty of moving larvae to change habitats [17]. Larvicides are a key tool, and increasing research is now focusing on natural essential oils, which show strong larvicidal activity and are biodegradable, making them an important potential source of larvicides [18]. The present findings showed that both perilla essential oil and 2-hexanoylfuran had good larvicidal activity, with 2-hexanoylfuran having a stronger larvicidal effect at the same concentration. The WHO does not clearly specify a value for the effective larvicidal concentration of reagents, but the default is that LC50 < 100 mg/L is active and LC50 < 50 mg/L is significant [19]. On the basis of these criteria, our perilla essential oil and 2-hexanoylfuran may be considered highly active and as potential sources of phytogenic larvicides.

Being a mixture of components, essential oils are complex and may have multiple larvicidal mechanisms acting in combination that have not been elucidated [19]. Some studies have demonstrated that essential oils inhibit acetylcholinesterase, causing neurotoxicity and leading to larval mortality [20]. In the present study, we measured the acetylcholinesterase activity of larvae after treatment with LC50 of perilla essential oil and 2-hexanoylfuran but found no change (Appendix A). However, we observed certain changes in the midgut associated with toxicity. The midgut epithelium has various functions such as digestion and absorption of nutrients, ionic- and osmoregulation, control of intestinal luminal pH, and secretion of digestive enzymes [21]. The histological sections of the midgut of larvae treated with perilla essential oil and 2-hexanoylfuran for 24 h showed a certain extent of damage. Changes in the LC50 group showed membrane rupture and cellular brush border damage, whereas the LC90 group showed a highly specialized damage variation with numerous severely vacuolated cells implying death. Our results are consistent with those reported in the midgut cells of *Aedes aegypti* exposed to acetogenin [22], indicating that perilla essential oil and 2-hexanoylfuran act by disrupting midgut function to affect larval growth and development. Moreover, we observed various forms of deformities in dead larvae, which were consistent with those reported in larvae treated with red seaweed extract; it is speculated that this effect may be attributed to the penetration of lipophilic compounds into the cuticle [23]. Furthermore, some of the essential oil-treated larvae developed into larval–pupal intermediates and pupal–adult intermediates, indicating that perilla essential oil has growth inhibitory effects that are likely associated with chitin synthesis and damage to the midgut [24].

Early and timely control of egg hatching before larval emergence is an effective method of vector control [25]. In general, *Culex* eggs are aggregated, and the chorionic membrane increases in hardness over time during embryonic development to resist unfavorable conditions such as dryness and extreme temperatures [26]. Therefore, to control the proliferation of mosquitoes, treatment with ovicides at the early stages of egg production is the optimal choice [27]. Our results showed that perilla essential oil and 2-hexanoylfuran could achieve high ovicidal effect, with 2-hexanoylfuran being more effective at the same concentration. Notably, at the same concentration, perilla essential oil is more effective than the water-soluble *Moringa oleifera* lectin (WSMoL), an emerging ovicide that is currently being studied in preventing eggs from hatching [28]. Therefore, perilla essential oil with 2-hexanoylfuran may be considered as a potential source of phytogenic ovicides.

In general, essential oils tend to exhibit a combination of multiple modes of action owing to the presence of multiple compounds and exhibit better toxicity than the monomers alone [29]. However, this was not observed in the larvicidal, ovicidal and fecundity assays in the present study. Therefore, it can be tentatively determined that the multiple components of perilla essential oil do not act enhanced when used in the water column, and there may be some antagonistic effect that renders it less toxic than a single major component [30]; however, this needs to be further investigated.

Repellent used to avoid mosquito bites is a key approach to interrupt the occurrence of mosquito-borne diseases [31]. DEET, the most commonly used commercial repellent in the market, has been reported to cause adverse reactions such as dermatitis, allergy, and even neurotoxicity [32]. Therefore, there is a need to promote novel mosquito repellents as alternatives to meet the multiple needs of the public. In particular, plant-derived extracts are an important source for the development of novel mosquito repellents owing to their efficacy and safety in mammals [33]. Our results showed that both perilla essential oil and 2-hexanoylfuran showed 100% mosquito repellency at the initial stage of application. However, 1% or 5% 2-hexanoylfuran became ineffective in 30–60 min. After 360 min, the positive control, DEET, showed a tendency of reduced effectiveness, but 10% perilla essential oil showed better repellent activity, which is consistent with the finding that a compound isolated from *Zingiber cassumunar* essential oil acts as a mosquito repellent [34]. We speculate that this observation may be associated with the high volatility of 2-hexanoylfuran, and the essential oil as a mixture has some advantage as a repellent [35]. However, future research needs to investigate how the effective duration of mosquito repellents on human skin may be increased without repeated application.

Oviposition is an important stage in the life cycle of mosquitoes, and oviposition deterrent activity is important to reduce the establishment of mosquito breeding sites [36]. Synthetic insecticides have been reported to have no effect on the selection of mosquito egg-laying sites [37], but many plant extracts have shown oviposition deterrent activity against mosquitoes [38]. Our results showed that both perilla essential oil and 2-hexanoylfuran had high oviposition deterrent effects at very low concentrations. Therefore, perilla essential oil and other plant-derived extracts may be used as oviposition deterrents, which offer a potentially effective and environmentally friendly solution to eliminate mosquito breeding sites.

The effect of pesticides on nontarget species is a key aspect to consider before their application. Notably, both essential oil and 2-hexanoylfuran exhibited some toxic effects on certain nontarget aquatic organisms, which may limit their usage in the water column. However, studies have shown that changing the formulation of essential oils reduces its effects on nontarget aquatic organisms [39,40]. We are continuing to work on it. However, perilla essential oil and 2-hexanoylfuran showed no toxic effect to human cells, and volunteers reported no discomfort when they were applied to the skin, indicating that perilla essential oil and 2-hexanoylfuran may be used as an effective mosquito repellent.

Thus, in the present study, the essential oil extracted from the traditional medicinal plant *Perilla frutescens* was analyzed to demonstrate that the main component is 2-hexanoylfuran. Both perilla essential oil and 2-hexanoylfuran showed good larvicidal effects, ovicidal effects, repellency, and oviposition deterrence in *Culex pipiens pallens*, suggesting that perilla essential oil and 2-hexanoylfuran are highly promising candidates for insecticides of plant origin.

## 4. Materials and Methods

### 4.1. Mosquitoes

The larvae of *Culex pipiens pallens* used in this experiment were collected from the fields of Tangkou Village (Shandong Province, China) and reared under standard laboratory conditions at 28 ± 1 °C and 70–80% humidity with a photoperiod of 12:12 h. The larvae were fed a yeast-rich diet in the laboratory, whereas adult mosquitoes were reared in cages and fed 5% glucose water daily for 3–4 days after emerging. Then, the females were supplied fresh mouse blood to promote reproduction [41,42].

### 4.2. Plant Material and Extraction

Perilla leaves were obtained from a commercial herbal store (Daoyuantang, Jiangsu, China). The plant was identified, and the samples were stored in the Department of Pathogen Biology at Nanjing Medical University. Approximately 100 g of air-dried perilla leaves was crushed and subjected to water distillation for approximately 2 h in a 1 L flask containing 500 mL of water using a Clevenger apparatus. The essential oil yield can be increased to 1.2% by adding 5% NaCl during distillation [43]. The oil layer was separated into brown glass bottles, dried with anhydrous Na_2_SO_4_, and stored at 4 °C away from light until the samples were analyzed by gas chromatography–mass spectrometry (GC-MS).

### 4.3. Gas Chromatography–Mass Spectrometry Analysis of Essential Oil

In this study, the quantitative and qualitative analysis of the essential oil was performed using GC-MS03 (Agilent-Technologies, Palo Alto, CA, USA) equipped with a TG-5SILMS fused silica capillary column (30 m × 0.25 mm inner diameter, 0.25 µm film thickness). The oven temperature programs were followed to reach 50 °C for 3 min; the temperature was raised to 100 °C (10 °C/min) and then from 280 °C (4 min, 15 °C/min) to 320 °C (8 min, 30 °C/min). The comparison of retention indices between compounds, mass spectra with computer data banks (NIST), and published data were used to identify the compounds.

### 4.4. Larvicidal Activity Bioassay

Larvicidal bioassays were performed according to the standard methods recommended by the WHO [44]. Perilla essential oil and its main component, 2-hexanoylfuran (Thermo Fisher Scientific, Waltham, MA, USA), were dissolved in 1 mL of ethanol, respectively, and then diluted with dechlorinated water to obtain 100 mL of different test concentrations. Perilla essential oil was assayed at 35, 40, 45, 50, 55, and 60 mg/L, whereas 2-hexanoylfuran was assayed at 15, 20, 25, 30, 35, and 40 mg/L. The control solution was prepared with 1 mL of ethanol in 99 mL of dechlorinated water. Using a Pasteur pipette, 25 third-instar larvae per cup were transferred into a cup containing 100 mL of the test solution. Each test concentration was tested three times. No food was provided to the larvae during the bioassay [45]. Dead larvae were counted after 24 h of exposure to determine toxicity of perilla essential oil and 2-hexanoylfuran, and the morphology of dead larvae was observed. Probit analysis was used to determine the larval lethal concentration (LC50). A small amount of yeast-rich foods was provided to the surviving larvae to observe growth-disrupting effects such as formation of intermediates.

### 4.5. Ovicidal Activity Bioassay

Ovicidal activity was determined using freshly laid eggs [46]. The concentrations of the solutions were configured in the same manner as in the larvicidal assay. Twenty-four eggs per group were placed in different concentrations of 70, 80, 90, 100, 110, and 120 mg/L of perilla essential oil and 40, 50, 60, 70, and 80 mg/L of 2-hexanoylfuran. The control solution was dechlorinated water containing ethanol. Three replicates were used per group. After 120 h of treatment, the eggs were microscopically observed to assess ovicidal activity of perilla essential oil and 2-hexanoylfuran.

### 4.6. Repellency Bioassay

This experiment was conducted in accordance with the WHO guidelines for efficacy testing [47]. Approximately 50–100 nonblood-feeding female mosquitoes (4–7 days old) were placed in a 20 × 20 × 30 cm^3^ mosquito cage and starved for 24 h before the test, which was conducted from 16:00 to 24:00 h. Four volunteers, two males and two females, were recruited and were required to have had no contact with hand sanitizer and soap on the day of the experiment. Volunteers wore rubber gloves on both hands and exposed 4 × 4 cm^2^ (16 cm^2^) of skin on the back of their hands. Repellency bioassay was first performed by applying 1.5 µL/cm^2^ (24 µL) diluent (ethanol) to the left hand; after letting the diluent dry for 1 min, the left hand was placed inside the mosquito cage for 30 s. The number of mosquitoes that landed on and/or started to pierce the skin was counted; if the number was greater than 10, the experiment was continued beyond the first 30 s. The left hand was coated with 24 µL of 10% *N,N*-diethyl-m-toluamide (DEET) as positive control and the right hand was coated with 24 µL of the repellent to be tested at different concentrations (1%, 5%, and 10%). Both hands were placed in the cage every 30 min for 30 s for 6 h, and the number of mosquitoes that landed on and/or started to pierce the skin was counted. The experiment was concluded when the repellency was below 50%.
$$Repellency\;\%=\frac{NC-NT}{NC}\;\times \;100$$
where NC is the number of mosquitoes in the control group and NT is the number of mosquitoes in each test group.

### 4.7. Oviposition Assay

Pregnant female mosquitoes (*n* = 120; 5–7 days old) were placed in 30 × 20 × 20 cm^3^ mosquito cages 48 h after blood feeding and provided with 5% glucose solution. Two plastic cups containing perilla essential oil (20, 40, 60, 80, 100 mg/L) or 2-hexanoylfuran (10, 15, 20, 25, 30 mg/L) at the same concentrations were placed in the cages, and two plastic cups containing an equal volume of dechlorinated water and ethanol, respectively, were placed as controls. The position of the cups was changed between replicates to eliminate any effect of position on oviposition. For each concentration, the experiment was performed in triplicate, and each bioassay was placed side by side in cages. After 48 h, the number of egg rafts in the treatment and control cups was observed and recorded. Effective repellency (ER%) and the oviposition activity index (OAI) were calculated for each concentration. Solutions with an OAI value of +0.3 or higher was considered an attractant, whereas solutions with an OAI value of −0.3 or lower were considered repellents [48].
$$\mathrm{ER}\%=\frac{\left(NC-NT\right)}{NC}\;\times \;100\%\;\;\;\;\;\;\;\;\;\;\;\;\mathrm{OAI}=\frac{\left(Nt-NC\right)}{\left(NT+NC\right)}$$
where NT is the total eggs laid in the extract solution and NC is the total eggs laid in the control solution.

### 4.8. Gut Histopathology

Third-instar larvae were exposed to LC50 and LC90 concentrations of perilla essential oil and 2-hexanoylfuran for 24 h, respectively, and controls were maintained for the same period without treatment. The larvae were fixed in 4% paraformaldehyde for 24 h. Subsequently, the samples were dehydrated in a graded ethanol series and embedded in tissue paraffin. The paraffin-embedded tissue blocks were sectioned into 4 µM thick sections using a microtome, stained with hematoxylin and eosin, and then viewed using an AXIO fluorescence microscope (Zeiss, Jena, Germany).

### 4.9. Effects on Nontarget Organisms

We used zebrafish and human embryonic kidney cells (293T) to assess toxicity to nontarget organisms. Zebrafish *(Danio rerio*) were kept at 24 °C water temperature with a photoperiod of 12:12 h. They were fed with an appropriate amount of yeast-rich foods. Ten individuals each were allocated to treatment groups with different concentrations (LC50, LC90) of perilla essential oil and 2-hexanoylfuran and control group with dechlorinated tap water, and mortality was recorded after 24 h. There were three replicates for each concentration.

A cell cytotoxicity assay was performed using the cell counting kit-8 assay (CCK-8; Beierbo, Nanjing, China). A suspension of 293T cells was added to 96-well plates (100 µL per well), and the plates were incubated for 24 h at 37 °C and 5% CO_2_ [42]. Dimethyl sulfoxide (DMSO; Sigma, St. Louis, MI, USA) at 0.1% (*v/v*) was used to dissolve perilla essential oil and 2-hexanoylfuran. After adding perilla essential oil (up to 45 and 58 mg/L) or 2-hexanoylfuran (up to 25 and 35 mg/L) per well, the plate was further incubated for 1 h; then, 10 µL of CCK-8 solution was added to each well, and the absorbance at 450 nm was measured after 1 h of incubation. Ten replicate wells were used for each concentration.

### 4.10. Statistical Analysis

The LC50 of perilla essential oil was calculated on the basis of the number of dead larvae. It was calculated by probit analysis with a dependability interval of 95% using SPSS. The dosage effect was determined to be significant for all adjusted models if the *P*-value was <0.05. One-way analysis of variance (ANOVA), followed by Duncan’s multiple range test (*p* < 0.001), was employed to investigate ovicidal and oviposition data. Analyses were performed using GraphPad Prism 8.0 software and SPSS version 27, and a *p*-value < 0.05 was considered to indicate a statistically significant difference.

## 5. Conclusions

The present study validates the effectiveness of perilla for mosquito control at different life stages (larvae, eggs, and adults) on the basis of its traditional usage, and demonstrates that perilla essential oil may be used as a larvicide, ovicide, repellent, and oviposition deterrent. Our results suggest that perilla essential oil and its main component are potential candidates for phytogenic insecticides that can be used as an effective mosquito population reduction strategy for controlling mosquito-borne diseases. In the future, our work will focus on developing nanoforms for field applications.

## Figures and Tables

**Figure 1 plants-12-01516-f001:**
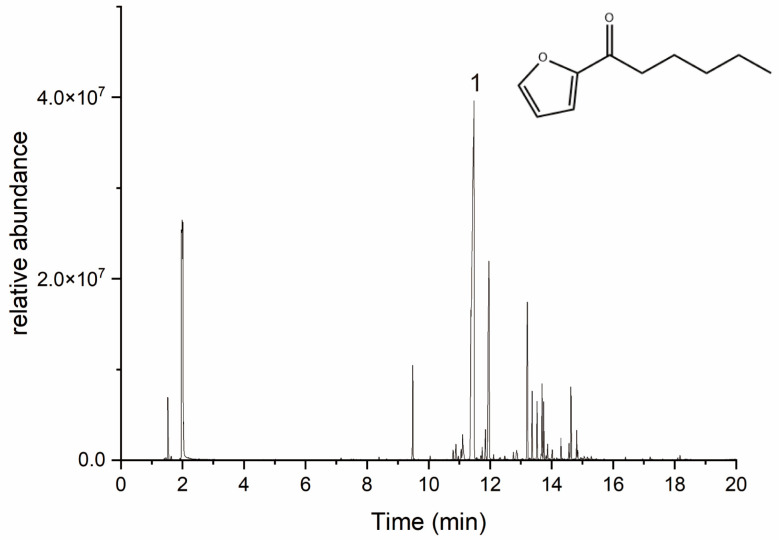
Gas chromatography–mass spectrometry chromatogram of perilla essential oil and the structure of the main component 2-hexanoylfuran.

**Figure 2 plants-12-01516-f002:**
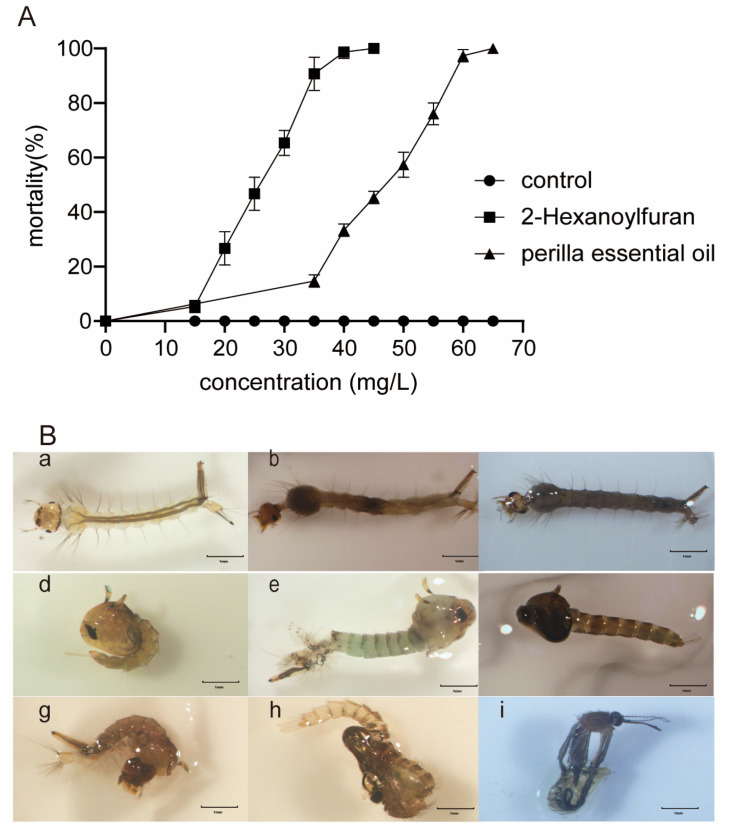
Larvicidal activity of *Perilla frutescens* essential oil (PE-EO) and 2-hexanoylfuran against *Culex pipiens pallens*. (**A**) Larval mortality of third-instar larvae of *Cx. pipiens pallens* post treatment with PE-EO and 2-hexanoylfuran. (**B**) Photographs showing morphological deformities and growth-disruptive symptoms in *Cx. pipiens pallens*: (**a**) a healthy larva from control; (**b**,**c**) dead larvae from treatment group; (**d**) a healthy pupa from control; (**e**) larvae with malformation; (**f**) dead black pupa from treatment group; (**g**) larval–pupal intermediates; (**h**) pupal–adult intermediates; (**i**) failed emerging adults.

**Figure 3 plants-12-01516-f003:**
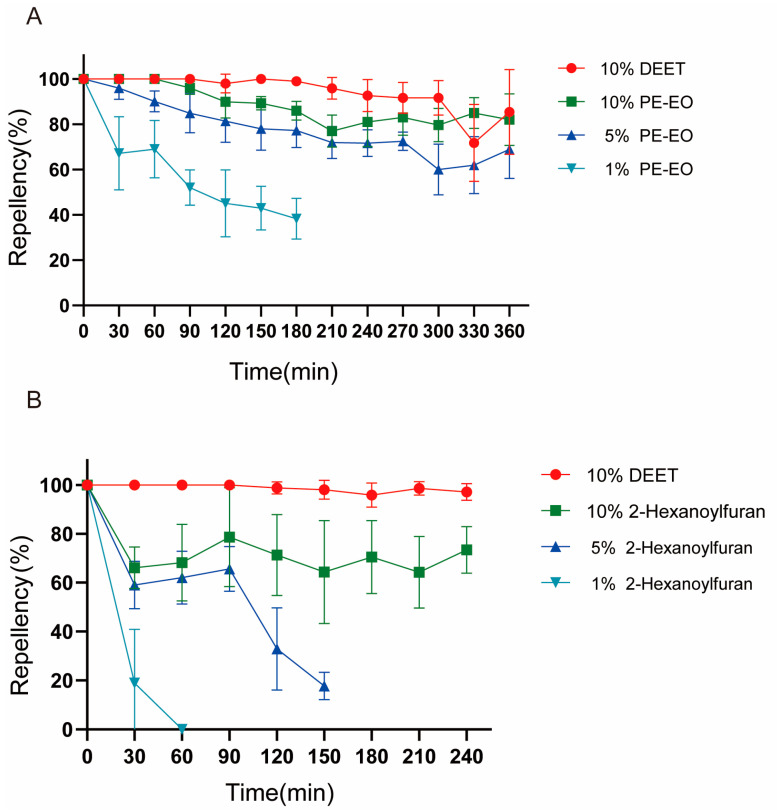
Repellency of perilla essential oil (**A**) and 2-hexanoylfuran (**B**) against *Culex pipiens pallens* adults.

**Figure 4 plants-12-01516-f004:**
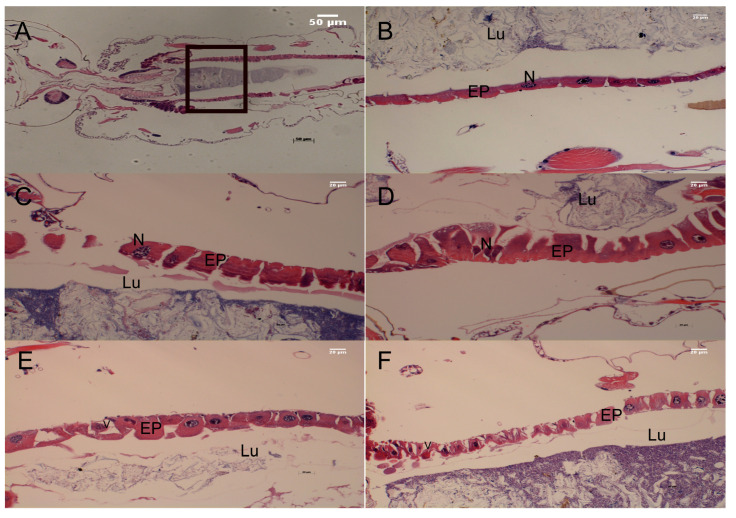
Photomicrographs of the midgut of *Culex pipiens pallens* third-instar larvae (magnification 20×). (**A**,**B**) control; (**C**,**D**) from a larva treated with *Perilla frutescens* essential oil (PE-EO; LC50, LC90); (**E**,**F**) from a larva treated with 2-hexanoylfuran (LC50, LC90). EP: Epithelial cells, N: nucleus, LU: gut lumen, V: vacuoles.

**Table 1 plants-12-01516-t001:** Chemical components of the essential oil extracted from *Perilla frutescens* leaves.

Peak No.	RT Time/Min	Peak Area (%)	Compound Name	CAS Number
1	7.157	0.11%	Benzaldehyde	100-52-7
2	8.391	0.12%	(+)-(R)-limonene	5989-27-5
3	9.486	3.64%	Linalool	78-70-6
4	10.802	0.42%	3-Methyl-1-(3-methyl-2-furyl)-1-butanone	488-05-1
5	10.892	0.63%	1-Propanone, 1-(3-cyclohexen-1-yl)-2,2-dimethyl-	16076-65-6
6	11.066	0.40%	1H-Indene, 1-ethylideneoctahydro-7a-methyl-, (1E,3aR,7aS)-rel-	56324-68-6
7	11.477	49.77%	2-hexanoylfuran	14360-50-0
8	11.706	0.20%	Cyclohexanemethanol, 4-(1-methylethenyl)-, trans-	22451-48-5
9	11.747	0.44%	dl-Perillaldehyde	2111-75-3
10	11.964	14.13%	Naphthalene, decahydro-1-methyl-2-methylene-	90548-09-7
11	12.118	0.20%	Methylgeranate	2349-14-6
12	12.481	0.20%	Eugenol	97-53-0
13	12.865	0.65%	β-Elemene	515-13-9
14	13.214	7.64%	β-Caryophyllene	87-44-5
15	13.368	2.18%	(E)-β-Farnesene	18794-84-8
16	13.53	1.95%	α-Caryophyllene	6753-98-6
17	13.661	0.21%	β-Ionone	14901-07-6
18	13.692	2.40%	(Z,E)- α-farnesene	26560-14-5
19	13.746	2.10%	β-Cubebene	13744-15-5
20	13.821	0.19%	(E,E)-α-Farnesene	502-61-4
21	13.872	0.60%	1-methyl-4-(1-methylethylidene)-2-(1-methylvinyl)-1-vinylcyclohexane	3242-08-8
22	14.024	0.47%	beta-Cadinene	523-47-7
23	14.306	0.65%	(±)-trans-Nerolidol	40716-66-3
24	14.569	0.63%	1H-Cycloprop[e]azulen-7-ol, decahydro-1,1,7-trimethyl-4-methylene-, (1aS,4aS,7R,7aS,7bS)-	77171-55-2
25	14.632	2.68%	(−)-Caryophyllene oxide	1139-30-6
26	14.816	0.99%	Apiole	523-80-8
27	14.852	0.33%	humulene epoxide ii	19888-34-7
28	15.168	0.19%	α-cadinol	481-34-5
29	17.205	0.16%	Palmitic acid	57-10-3
30	18.175	0.16%	Phytol	150-86-7
	Total	94.42%		

**Table 2 plants-12-01516-t002:** Ovicidal activity of Perilla essential oil (PE-EO) and 2-hexanoylfuran against *Culex pipiens pallens*.

Treatment	Concentration (mg/L)	Egg Hatchability (%)	F Value	*p* Value
control	-	100 ± 0.0	-	-
PE-EO	70	90.2 ± 2.4 a	383.9	<0.001
	80	65.2 ± 2.4 b		
	90	41.7 ± 4.2 c		
	100	19.4 ± 6.5 d		
	110	15.3 ± 2.4 d		
	120	1.30 ± 2.4 e		
2-Hexanoylfuran	40	86.1 ± 2.4 a	153.21	<0.001
	50	62.5 ± 4.2 b		
	60	56.9 ± 2.4 b		
	70	23.6 ± 8.7 c		
	80	1.38 ± 2.4 d		

Means within a row followed by different letters indicate statistically significant differences (ANOVA and Duncan’s multiple range test, *p* < 0.001).

**Table 3 plants-12-01516-t003:** Oviposition activity of perilla essential oil (PE-EO) and 2-hexanoylfuran against *Culex pipiens pallens*.

Treatment	Concentration (mg/L)	ER%	OAI	Effect (A/N/R)
PE-EO	20	49.4 ± 12 c	−0.33	R
	40	81.3 ± 1.8 b	−0.69	R
	60	87.7 ± 1.5 ab	−0.78	R
	80	90.0 ± 1.0 ab	−0.82	R
	100	94.7 ± 2.7 a	−0.89	R
2-Hexanoylfuran	10	62.6 ± 2.1 d	−0.46	R
	15	72.1 ± 5.6 c	−0.57	R
	20	85.1 ± 0.7 b	−0.74	R
	25	88.1 ± 1.5 ab	−0.79	R
	30	93.1 ± 2.3 a	−0.87	R

*ER*: effective repellency, *OAI*: oviposition active index, *Effect: A*: attractant, *N*: no effect, *R*: repellent. Means within a row followed by different letters indicate statistically significant differences (ANOVA and Duncan’s multiple range test, *p* < 0.001).

## Data Availability

The data is contained within the manuscript and Appendix A.

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
