# Peer review of "Toxic Effects of Perilla frutescens (L.) Britt. Essential Oil and Its Main Component on Culex pipiens pallens (Diptera: Culicidae)"

_plants, 2023, doi:10.3390/plants12071516_

Round 1

Reviewer 1 Report

Quality publication that should be published. Detailed comments are attached.

Author Response

Thank you very much for your time involved in reviewing the manuscript and your very encouraging comments on the merits. The specific response is in the attachment

Reviewer 2 Report

Journal Plants (ISSN 2223-7747)

Manuscript ID plants-2253180

Type Article

Title  Toxic effects of Perilla frutescens (L.) Britt. essential oil and its main component on Culex pipiens pallens (Diptera: Culicidae)

Authors  Ruimin Zhang , Wenxing Zhang , Junnan Zheng , Jingwei Xu , Huan Wang , Jiajia Du , Yan Sun , Dan Zhou , Bo Shen 

Section  Phytochemistry

Special Issue  Phytochemical Composition and Biological Activity

Abstract   This study aimed to analyze the main components of Perilla frutescens essential oil (PE-EO), investigate the specific activity of PE-EO as a botanical insecticide and mosquito repellent, and explore whether its main constituents are potential candidates for further research. The larvicidal activity assay showed that LC50 of PE-EO and 2-hexanoylfuran was 45 and 25 mg/L, respectively. In the ovicidal activity assay, both 120 mg/L PE-EO and 80 mg/L 2-hexanoylfuran could achieved 98% egg mortality. Moreover, PE-EO and 2-hexanoylfuran showed repellency and oviposition deterrence effects. Notably, 10% PE-EO maintained a high rate of protection for 360 min. Therefore, perilla essential oil is an effective agent for mosquito control at several life stages and that its main component, 2-hexanoylfuran, is a potential candidate for developing novel plant biopesticides.

Concerns:

I was concerned about your GC-MS identification of main constituent of P. frutescens essential oil (PE-EO) as 2-hexanoylfuran (49%) CAS 14360-50-0. 

 images of 2 cpds in attached file:   2-hexanoylfuran         perilla ketone

You cited article by Tian et al. (2014) that stated predominant components P. frutescens essential oils in China were 2-acetylfuran (max. 82.17%), perillaldehyde (max. 53.41%), caryophyllene (max. 38.34%), laurolene (max. 40.6%), 2-hexanoylfuran (max. 33.03%)

Verma et al. (2015) stated the main constituent of the essential oil of P. frutescens was identified as perilla ketone (48.6%) CAS 553-84-4 =  1-(Furan-3-yl)-4-methylpentan-1-one 

Eldeghedy et al. (2022) stated the main constituent of the essential oil of P. frutescens was identified as L-perillaldehyde

Suggestions:

I found very little to change in the text.

Line 316 spelling:  …chitin synthesis

Author Response

(The authors gave the same response as above.)
